# Understanding and Enhancing Safety Mechanisms of LLMs via Safety-Specific Neuron

**Yiran Zhao**[1][†]   **Wenxuan Zhang**[2]   **Yuxi Xie**[1]   **Anirudh Goyal**[3]
**Kenji Kawaguchi**[1]   **Michael Qizhe Shieh**[1][†]
[1] National University of Singapore   [2] Singapore University of Technology and Design
[3] Google DeepMind

## Abstract

Safety alignment for large language models (LLMs) has become a critical issue due to their rapid progress. However, our understanding of effective safety mechanisms in LLMs remains limited, leading to safety alignment training that mainly focuses on improving optimization, data-level enhancement, or adding extra structures to intentionally block harmful outputs. To address this gap, we develop a neuron detection method to identify safety neurons—those consistently crucial for handling and defending against harmful queries. Our findings reveal that these safety neurons constitute less than $1\%$ of all parameters, are language-specific and are predominantly located in self-attention layers. Moreover, safety is collectively managed by these neurons in the first several layers. Based on these observations, we introduce a $\underline{S}$afety $\underline{N}$euron $\underline{Tun}$ing method, named `SN-Tune`, that exclusively tune safety neurons without compromising models' general capabilities. `SN-Tune` significantly enhances the safety of instruction-tuned models, notably reducing the harmful scores of Llama3-8B-Instruction from 65.5 to 2.0, Mistral-7B-Instruct-v0.2 from 70.8 to 4.5, and Vicuna-13B-1.5 from 93.5 to 3.0. Moreover, `SN-Tune` can be applied to base models on efficiently establishing LLMs' safety mechanism. In addition, we propose $\underline{R}$obust $\underline{S}$afety $\underline{N}$euron $\underline{Tun}$ing method (`RSN-Tune`), which preserves the integrity of LLMs' safety mechanisms during downstream task fine-tuning by separating the safety neurons from models' foundation neurons.[1]

## 1 Introduction

The rapid developments of large language models (LLMs) (Achiam et al., 2023; Jiang et al., 2023; Reid et al., 2024; Team et al., 2024; Dubey et al., 2024) have brought safety alignment to the forefront of research (Zou et al., 2023; Zhao et al., 2024d; Zou et al., 2024; Deng et al., 2024; Wei et al., 2024a). Different perspectives have been studied to improve safety alignments, such as improving optimization (Ouyang et al., 2022; Rafailov et al., 2024; Yuan et al., 2023), refining training data (Zhou et al., 2024; Rafailov et al., 2024; Zhang et al., 2024), or implementing additional structures designed to intentionally block harmful outputs (Inan et al., 2023; Zou et al., 2024). Despite its importance, a clear understanding of safety mechanisms in LLMs remains absent. Prior works tried to identify and interpret safety mechanisms in LLMs from either layer-level (Li et al., 2024) or feature-level (Chen et al., 2024). However, their identification methods attribute nearly $10\%$ of parameters to safety-related functions. This large proportion makes it challenging to effectively perform safety alignments based on these findings (Anwar et al., 2024; Zeng et al., 2024). Moreover, other works have suggested that safety mechanisms can be easily compromised through minor parameter adjustments (Qi et al., 2024; Zhao et al., 2024a).

In this work, we aim to understand and interpret safety mechanisms in LLMs at a finer granularity, specifically at the neuron level across all structures, including the self-attention and feed-forward parts. Here, a "neuron" is represented by a single row or column of a parameter matrix in LLMs. We identify a "safety neuron" as one that consistently plays a crucial role in processing and defending against harmful queries. Specifically, a neuron is considered important if its removal—by setting

---

[†]Correspondence to: Yiran Zhao (zhaoyiran@u.nus.edu), Michael Shieh (michaelshieh@comp.nus.edu.sg).

[1]Our code is publicly available at `https://github.com/zhaoyiran924/Safety-Neuron`.

its parameters to zero—significantly affects the generated output beyond a specified threshold. To achieve this, we input a corpus of harmful queries and extract neurons that are important across all queries in the corpus, identifying them as the set of safety neurons in the LLM. By conducting a thorough analysis of these identified safety neurons in various models, we uncover several key insights about LLMs' safety mechanisms: First, we find that safety neurons comprise less than $1\%$ of all parameters. Second, each language has its own unique safety neurons, with minimal overlap between them. Third, safety is collaboratively managed by safety neurons located in the first several layers of the model. Lastly, safety neurons are predominantly located within the self-attention structures.

Motivated by these intriguing observations, we propose a Safety Neuron Tuning method (SN-Tune), designed to exclusively tune the safety neurons in LLMs. As shown in Figure 1, we gather safety training documents that include harmful queries and refusal safety outputs, similar to the widely used safety alignment training settings (Inan et al., 2023; Zhang et al., 2024; Zou et al., 2024). We then tune the identified safety neurons while leaving other safety-unrelated neurons unchanged by setting their gradients to zero during the tuning process. Experimental results demonstrate that SN-Tune not only enhances the safety mechanism for instruction-tuned models but also establishes safety mechanism for base models without compromising their general capabilities.

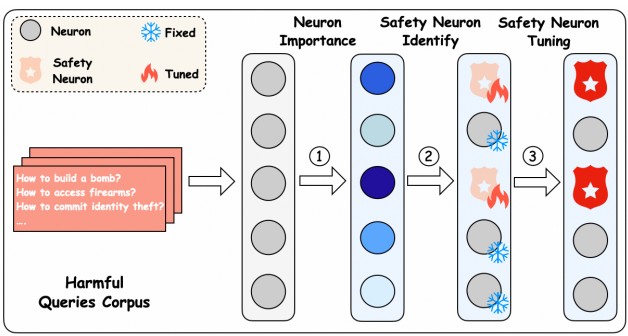

Figure 1: SN-Tune mainly consists of three steps: ① calculating neuron importance for handling harmful queries; ② identifying "safety neuron" that consistently play a crucial role in processing harmful queries; ③ tune the identified safety neurons while leaving other safety-unrelated neurons unchanged during the tuning process.

Notably, it reduces the average harmful scores of Llama3-8B-Instruction from 65.5 to 2.0, Mistral-7B-Instruct-v0.2 from 70.8 to 4.5, and Vicuna-13B-1.5 from 93.5 to 3.0. Moreover, SN-Tune reduces base models' harmful score from around 100 to 5.3, 13.5, and 13.8 for LLama2-7B-Base, LLama3-8B-Base, and Mistral-7B-v0.1, respectively. The harmful score is evaluated using the harmful behavior dataset (Zou et al., 2023), by averaging the Attack Success Rate (ASR) across various adversarial attacking methods, including Direct Attack, GCG (Zou et al., 2023), AutoDAN (Liu et al., 2024) and PAIR (Chao et al., 2023). Concurrently, we assess the models' general capabilities using representative NLP tasks including MMLU (Hendrycks et al., 2020), ARC-Challenge (Clark et al., 2018), and GSM8K (Cobbe et al., 2021), ensuring that safety improvements do not come at the cost of overall performance.

Building upon the strong performance of SN-Tune, we aim to further enhance LLMs' safety robustness during downstream tasks fine-tuning, a common practice for users focusing on specific application scenarios (Yu et al., 2024; Zhao et al., 2024c). As Qi et al. (2024) observed, even fine-tuning with seemingly benign and widely used datasets can unintentionally compromise the safety alignment of LLMs. From the neuron perspective, fine-tuning on downstream tasks modifies certain foundation neurons (Zhao et al., 2024b; Liang et al., 2024). Consequently, the vulnerability of a model's safety mechanism to downstream task fine-tuning may be attributed to the overlap between these foundation neurons and safety neurons, with the latter being unintentionally adjusted during the fine-tuning process. Inspired by this observation, we propose another technique called Robust Safety Neuron Tuning method (RSN-Tune). It separates safety neurons from foundation neurons by selectively tuning only those safety neurons that do not overlap with foundation neurons when applying SN-Tune to instruction-tuned models. Experimental results demonstrate the effectiveness of RSN-Tune in enhancing models' safety robustness during downstream tuning. Notably, it reduces Llama2-7B-Chat's harmful score after tuning on GSM8K training set from 41.0 to 26.0 and Mistral-7B-Instruct-v0.2's from 79.0 to 41.0. Importantly, RSN-Tune enhances safety robustness while maintaining models' downstream tuning performance.

## 2 SAFETY NEURONS

In this section, we propose a neuron detection method that can calculate the importance of a neuron when handling a query without a corresponding labeled output.

### 2.1 SAFETY NEURON DETECTION

A *neuron* is defined as a single row or column of a parameter matrix in LLMs, including the self-attention and feed-forward structures. To identify neurons responsible for safety in an alignment-tuned LLM, it's crucial to extract those that play a key role in processing inputted harmful queries.

**Foundational Safety Neuron Detection**    Formally, we denote the $l$-th neuron in layer $i$ as $N_i^{(l)}$, while the intermediate representation after layer $i$ when handling harmful query $x$ is denoted as $h_i(x)$. Furthermore, the importance of neuron $N_i^{(l)}$ in processing $x$ is calculated by

$$\|h_{\setminus N_i^{(l)},i}(x) - h_i(x)\|_2, \tag{1}$$

where $h_{\setminus N_i^{(l)},i}(x)$ represents the intermediate representation after deactivating neuron $N_i^{(l)}$. Therefore, the activated neurons of the model when handling harmful query $x$ can be calculated by

$$\mathcal{N}_x = \{N_i^{(l)} | \|h_{\setminus N_i^{(l)},i}(x) - h_i(x)\|_2 \geq \epsilon, \text{ for all } N_i^{(l)} \text{ in LLM}\}, \tag{2}$$

where $\epsilon$ is a pre-defined threshold. Furthermore, after collecting a set of harmful queries, denoted as $X$. We extract neurons consistently activated for all queries in $X$, identifying the safety neurons we aim to obtain, i.e.,

$$\mathcal{N}_{\text{safe}} = \{N_i^{(l)} | N_i^{(l)} \in \mathcal{N}_x, \forall x \in X, \text{ for all } N_i^{(l)} \text{ in LLM}\}. \tag{3}$$

**Accelerated Safety Neuron Detection**    The process of deactivating $N_i^{(l)}$ sequentially in Equation 2 is extremely slow due to its sequential nature. Drawing inspiration from the parallel neuron detection method proposed by Zhao et al. (2024b), we implement it on safety neuron detection through the incorporation of masks and parallel computations. Specifically, for the feed-forward layer,

$$\|h_{\setminus N_i^{(l)},i}(x) - h_i(x)\|_2 = \|(h_{\text{ffn}}(x) \cdot \texttt{Mask})W_{down}\|_2, \tag{4}$$

where $h_{\text{ffn}}$ is the intermediate embedding between the up-projection and down-projection matrices, $\texttt{Mask}$ is an identity matrix of size $(\dim(h_{\text{ffn}}) \times \dim(h_{\text{ffn}}))$, and $W_{down}$ denotes the down-projection matrix in the feed-forward layer. Moreover, for the self-attention layer,

$$\|h_{\setminus N_i^{(l)},i}(x) - h_i(x)\|_2 \approx \left\|\text{softmax}\left(\frac{W_Q(x)W_K^T(x) - \Delta(x)}{\sqrt{d}}\right) - \text{softmax}\left(\frac{W_Q(x)W_K^T(x)}{\sqrt{d}}\right)\right\|_2, \tag{5}$$

where $W_Q$ and $W_K$ are the attention matrices for $Q$ and $K$, respectively, and $\sqrt{d}$ represents the corresponding dimension following the notations in Vaswani et al. (2017), and

$$\Delta(x) = W_Q(x).resize(l,1,d) \times W_K(x).resize(1,l,d) \in \mathbb{R}^{l \times l \times d}. \tag{6}$$

Detailed proof of Equation 4 and Equation 5 is available in Appendix A.1.

### 2.2 VERIFY IDENTIFIED SAFETY NEURON

We subsequently apply the accelerated safety neuron detection method to a variety of alignment-tuned LLMs to identify corresponding safety neurons, and conduct experiments to verify that these neurons are exclusively responsible for handling safety. Specifically, by deactivating the safety neurons, the model's safety mechanism will be attacked, potentially transforming it into a harmful model. However, by solely manipulating neurons associated with safety, the overall functionality should remain intact. Consequently, the model could become both helpful and harmful.

Table 1: Performance of models on harmfulness and general capability with the deactivation of safety neurons ("Deact-SN") and an equivalent number of randomly selected neurons ("Deact-R"). Harmfulness is measured by Attack Success Rate (lower is safer), and capability by Accuracy.

| Dataset | | Llama2-7B-Chat | | | Llama3-8B-Instruction | | | Mistral-7B-Instruct-v0.2 | | |
| --- | --- | --- | --- | --- | --- | --- | --- | --- | --- | --- |
| | | Origin. | Deact-R | Deact-SN | Origin. | Deact-R | Deact-SN | Origin. | Deact-R | Deact-SN |
| **Harmful↓** | Harm Behavior | 0.0 | 2.0 | 97.0 | 30.0 | 31.0 | 78.0 | 36.0 | 39.0 | 86.0 |
| | Adv Behavior | 0.0 | 3.0 | 83.0 | 7.0 | 13.0 | 96.0 | 30.0 | 30.0 | 87.0 |
| | MultiJail-En | 12.7 | 12.9 | 81.6 | 20.0 | 21.6 | 74.3 | 44.1 | 46.8 | 86.4 |
| | *Avg. Harmful* | 4.2 | 6.0 | **87.2** | 19.0 | 21.9 | **82.8** | 36.7 | 38.6 | **86.5** |
| **Capablity↑** | MMLU | 48.2 | 48.4 | 47.8 | 65.3 | 63.2 | 62.7 | 59.2 | 59.3 | 58.5 |
| | GSM8K | 24.8 | 22.7 | 21.9 | 75.9 | 73.6 | 72.4 | 43.6 | 43.6 | 42.1 |
| | *Avg. Capability* | **36.5** | 35.6 | 34.8 | **70.6** | 68.4 | 67.6 | 51.4 | **51.5** | 50.3 |

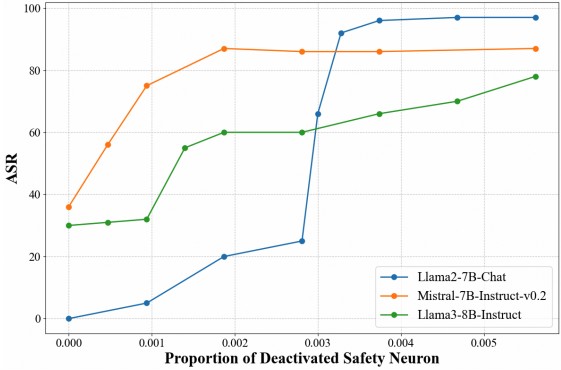

Figure 2: Effects of deactivated safety neurons on ASR.

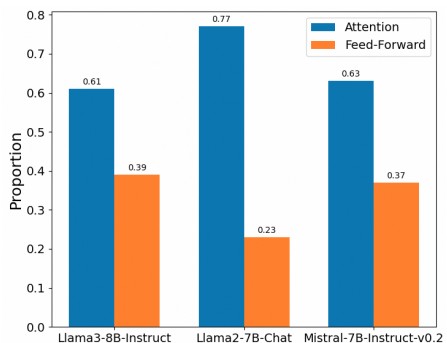

Figure 3: Distribution of Safety Neuron in different structures.

**Experimental Setup** We employ three open-source models that have been specifically tuned for safety, including Llama2-7B-Chat (Touvron et al., 2023), Llama3-8B-Instruction (Dubey et al., 2024), and Mistral-7B-Instruct-v0.2 (Jiang et al., 2023). The harmful corpus set used to detect safety neurons is constructed from the training set split in Zou et al. (2024). More details are illustrated in Appendix A.2. To prove the generability of the detected safety neuron, we test the harmfulness of the model on harmful behavior testset in Zou et al. (2023) (Harm Behavior), adversarial behavior testset in Mazeika et al. (2024) (Adv Behavior) and English version of multilingual jailbreak testset in Deng et al. (2024) (MultiJail-En). Furthermore, the models' general capability is evaluated by MMLU Hendrycks et al. (2020) and GSM8K Cobbe et al. (2021).

**Evaluation Metrics** The harmfulness is assessed through direct attacks using the Attack Success Rate (ASR), which identifies harmful keywords from the output, following the method outlined by Zou et al. (2023). Furthermore, accuracy is the metric used for MMLU and GSM8K.

**Existence of Safety Neurons** Table 1 demonstrates how deactivating safety neurons can attack the model's safety mechanism. Moreover, the model's general capabilities have not diminished, indicating that these neurons are specifically for safety mechanisms, not for other functions. Even with just about 0.5% of neurons deactivated, the model's safety capabilities are significantly compromised, leading to a substantial increase in harmful behavior: from 4.2 to 87.2 on Llama2-7B-chat, from 19.0 to 82.8 on Llama3-8B-Instruction, and from 36.7 to 86.5 on Mistral-7B-Instruct-v0.2. Meanwhile, randomly deactivating an equivalent number of neurons has little to no impact on the model's safety. Regarding general capability, deactivating the safety neuron shows minimal impact, similar to deactivating randomly selected neurons, as demonstrated by the performance of 36.5 and 34.8 on Llama2-7B-chat, 70.6 and 68.4 on Llama3-8B-Instruction, and 51.4 and 50.3 on Mistral-7B-Instruct-v0.2 before and after deactivation. Therefore, the detected neurons are safety neurons that are associated with safeguarding the models.

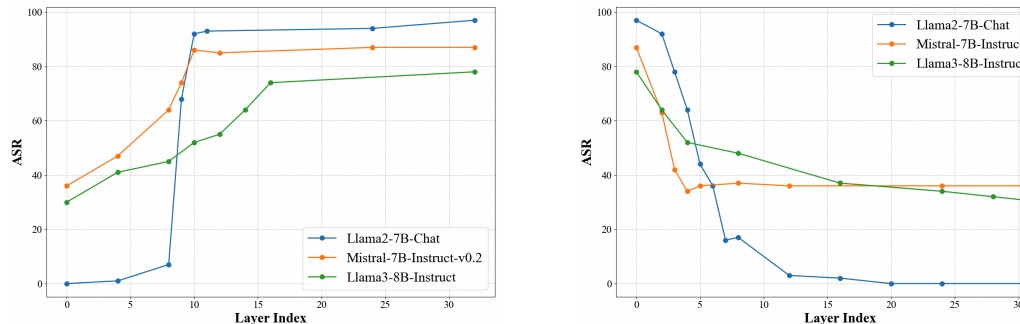

Figure 4: Effect of deactivating safety neurons in different layers. The left represents deactivating safety neurons *before* the certain layers, the right indicates deactivation *after* the certain layers.

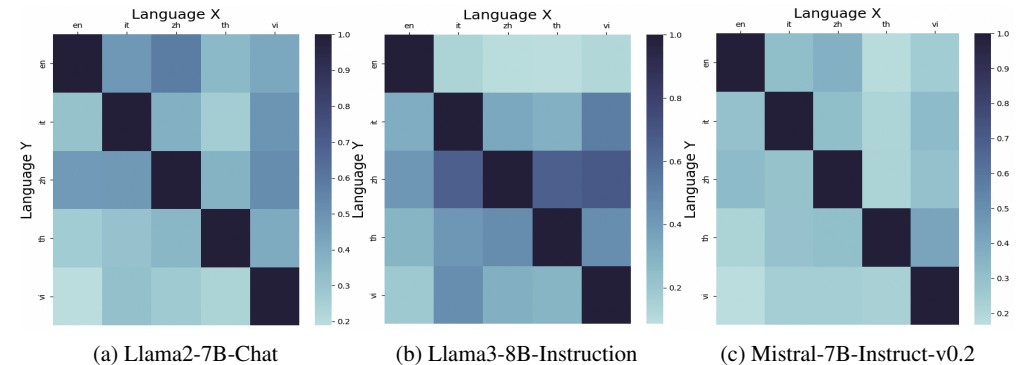

(a) Llama2-7B-Chat  (b) Llama3-8B-Instruction  (c) Mistral-7B-Instruct-v0.2

Figure 5: Overlapping ratio of safety neurons across different languages.

## 2.3 ANALYZE SAFETY MECHANISM IN LLMS

As we have detected the safety neurons of LLMs, we conduct a more detailed and comprehensive analysis of the properties of LLM's safety mechanism.

### 2.3.1 SAFETY MECHANISM PROPERTIES

**Safety mechanism is resilient but breakable by under one percent of the parameters.** Figure 2 shows the harmful score of three models as deactivating different number of safety neurons. In Mistral-7B-Instruct-v0.2, deactivating $0.2\%$ of neurons can destroy its safety mechanism, compared to $0.4\%$ for Llama2-7B-Chat and $0.5\%$ for Llama3-8B-Instruction. Furthermore, an emergence of "harmfulness" is observed for three models. For example, in Llama2-7B-Chat, the leap appears when deactivating $0.3\%$ neurons, while the number is $0.15\%$ for Llama3-8B-Instruction and is $0.1\%$ for Mistral-7B-Instruct-v0.2.

**Safety Mechanism is handled by the first several layers together.** Figure 4 illustrates the detrimental impact of deactivating safety neurons across various layers in models. Upon deactivating neurons in the first 10 layers simultaneously, we observe a near-complete breakdown in the safety mechanism of Llama2-7B-Chat. This threshold is 10 for Mistral-7B-Instruct-v0.2 and 16 for Llama3-8B-Instruction. On the contrary, if we deactivate safety neurons from the back to the front, the breakdown of safety mechanisms becomes apparent as nearly all safety neurons are deactivated.

**Safety neurons predominantly reside within the self-attention layers.** In Figure 3, safety neurons are categorized based on their belonging structures, which include the attention structure and feed-forward structure. Our findings reveal that safety neurons predominantly reside within the attention structure. Specifically, in Llama2-7B-Chat, $77\%$ of safety neurons are attributed to the attention structure, while $23\%$ belong to the feed-forward structure. This finding aligns with the interpretation that the attention structure primarily handles understanding, while the feed-forward structure is mainly

Table 2: Performance of `SN-Tune` on instruction-tuned models. General capabilities are evaluated by accuracy, while harmfulness is evaluated by ASR.

| Dataset | | Vicuna-13B-v1.5 | | | Llama3-8B-Instruction | | | Mistral-7B-Instruct-v0.2 | | |
|---|---|---|---|---|---|---|---|---|---|---|
| | | Origin. | Circ-Break | SN-Tune | Origin. | Circ-Break | SN-Tune | Origin. | Circ-Break | SN-Tune |
| **Training Cost** (min.) | | - | 43 | **4** | - | 24 | **2** | - | 23 | **2** |
| **# Parameters** (M) | | 0 | 34.1 | **0** | 0 | 27.5 | **0** | 0 | 27.5 | **0** |
| **Capablity**↑ | MMLU | 53.4 | 52.8 | **55.7** | 65.2 | 65.6 | **67.3** | 58.6 | 56.3 | **59.5** |
| | ARC-c | 59.7 | 61.3 | **61.6** | 73.7 | 74.1 | **74.9** | 72.6 | 71.8 | **73.4** |
| | GSM8K | 33.4 | **35.0** | 34.8 | 63.2 | 64.3 | **69.6** | 43.7 | 42.5 | **44.1** |
| | *Avg. Capablity* | 48.8 | 49.7 | **50.7** | 67.4 | 68.0 | **68.4** | 58.3 | 56.9 | **59.0** |
| **Harmful**↓ | Direct | 92.0 | 0.0 | **0.0** | 30.0 | 0.0 | **0.0** | 36.0 | 7.0 | **0.0** |
| | GCG | 100.0 | 3.0 | **0.0** | 74.0 | 3.0 | 4.0 | 88.0 | 8.0 | **6.0** |
| | AutoDAN | 93.0 | **2.0** | 3.0 | 82.0 | 0.0 | **0.0** | 91.0 | **3.0** | 4.0 |
| | PAIR | 89.0 | 16.0 | 9.0 | 76.0 | 9.0 | 4.0 | 68.0 | 22.0 | 8.0 |
| | *Avg. Harmful* | 93.5 | 5.3 | **3.0** | 65.5 | 3.0 | **2.0** | 70.8 | 10.0 | **4.5** |

responsible for knowledge extraction (Geva et al., 2021). Given that the safety mechanism focuses on understanding potential threats to discern their harmful nature without the need to extract much new knowledge, it is logical for safety neurons to predominantly reside in the attention structure, despite the attention parameters being fewer than half of the feed-forward parameters.

### 2.3.2 MULTILINGUAL SAFETY

Based on the research by Deng et al. (2024); Yong et al. (2024); Kotha et al. (2024), the safety mechanism cannot be effectively transferred between languages. For instance, even when a LLM is specifically tuned for safety in English, it may still pose risks when applied to other languages. Drawing inspiration from these discoveries, we analyze this phenomenon through the perspective of safety neurons. We specifically incorporate five languages—English (en), Italian (it), Chinese (zh), Thai (th), and Vietnamese (vi)—spanning high-resource to low-resource languages, to visualize the overlap of safety neurons. Specifically, the overlap among safety neurons are defined as $overlap(x, y) = |\mathcal{N}_x \cap \mathcal{N}_y|/|\mathcal{N}_y|$, where $\mathcal{N}_{\text{language}}$ represents the set of safety neurons in that language. Figure 5 displays the intersection of safety neurons across languages. Our analysis reveals that the overlap of safety neurons is typically below $30\%$, significantly less than that of language-specific neurons, which are a subset of neurons responsible for processing multilingual queries(Zhao et al., 2024b). This disparity underscores the unique nature of safety neurons in each language, indicating that safety capabilities are not transferrable between languages. This observation aligns with the progression of the SFT training, where diverse language-specific safety corpora are developed to provide tailored safety mechanism for individual languages (Zhang et al., 2024).

## 3 EFFICIENT SAFETY TRAINING

With only a limited number of parameters able to ensure safety, we can focus on manipulating these neurons effectively to strengthen or even establish the safety mechanism.

### 3.1 LIVE-LINE WORK ON INSTRUCT TUNED MODEL

**Experimental Setup** With fewer than $1\%$ of neurons dedicated to safety, we can enhance safety by fine-tuning them using a safety corpus, named as Safety Neuron Tuning (`SN-Tune`). Specifically, we create a safety corpus by partitioning a training dataset from (Zou et al., 2024), utilizing it to identify and strengthen safety neurons. In a manner similar to the setup in Table 1, we assess models' harmfulness using the harmful behavior testset, while their general capabilities are evaluated on MMLU (5-shots) (Hendrycks et al., 2020), ARC-c (3-shots) (Clark et al., 2018), and GSM8K (zero-shot) (Cobbe et al., 2021). Additionally, beyond testing direct attacks, we explore other attack methods, including GCG (Zou et al., 2023), AutoDAN (Liu et al., 2024), and PAIR (Chao et al., 2023). To demonstrate the generality of the method, we also employ the large model Vicuna-13B-v1.5 (Peng et al., 2023) in addition to Llama3-8B-Instruction and Mistral-7B-Instruct-v0.2. We

Table 3: Performance of `SN-Tune` on base models. General capabilities are evaluated by accuracy, while harmfulness is evaluated by ASR.

| Dataset | | Llama2-7B-Base | | | Llama3-8B-Base | | | Mistral-7B-v0.1 | | |
|---|---|---|---|---|---|---|---|---|---|---|
| | | Origin. | Circ-Break | SN-Tune | Origin. | Circ-Break | SN-Tune | Origin. | Circ-Break | SN-Tune |
| **Training Cost** (min.) | | - | 23 | 2 | - | 35 | 2 | - | 21 | 2 |
| **# Parameters** (M) | | 0 | 34.1 | 0 | 0 | 27.5 | 0 | 0 | 27.5 | 0 |
| **Capablity**↑ | MMLU | **49.2** | 49.1 | **49.2** | 70.1 | 68.9 | **69.6** | 68.4 | 68.1 | **69.2** |
| | ARC-c | 27.6 | 26.8 | **29.3** | 70.7 | **72.0** | 71.8 | **74.8** | 73.4 | 74.7 |
| | GSM8K | 12.7 | 13.7 | **16.3** | 58.9 | 58.2 | **59.5** | 50.4 | 47.6 | **52.3** |
| | *Avg. Capablity* | 29.8 | 29.9 | **31.6** | 66.6 | 66.4 | **67.0** | 62.0 | 63.0 | **65.4** |
| **Harmful**↓ | Direct | 97.0 | 84.0 | **0.0** | 100.0 | 87.0 | **0.0** | 100.0 | 78.0 | **6.0** |
| | GCG | 100.0 | 92.0 | **7.0** | 100.0 | 95.0 | **14.0** | 100.0 | 82.0 | **13.0** |
| | AutoDAN | 100.0 | 97.0 | **9.0** | 100.0 | 92.0 | **21.0** | 100.0 | 93.0 | **12.0** |
| | PAIR | 98.0 | 89.0 | **5.0** | 100.0 | 96.0 | **19.0** | 100.0 | 97.0 | **24.0** |
| | *Avg. Harmful* | 98.8 | 90.5 | **5.3** | 100.0 | 92.5 | **13.5** | 100.0 | 87.5 | **13.8** |

compare `SN-Tune` with Zou et al. (2024), who train an independent model called "Circ-Break" to act as a circuit breaker, interrupting models when they produce harmful outputs.

**Experiment Details** We utilize the HarmBench implementation (Mazeika et al., 2024) for the attacking methods. For general capability evaluation, we employ accuracy as the metric, while for harmfulness assessment, we use Attack Success Rate (ASR). The hyperparameters for fine-tuning primarily focus on the training corpus, number of epochs, and learning rate. As the fine-tuning process is essentially continued training, we aim to minimize alterations to the existing parameters. Specifically, we use a dataset of 50 documents where the model refuses to answer harmful questions, train for only 1 epoch, and set the initial learning rate to $1e-6$.

**Main Results** Table 2 shows the performance of `SN-Tune` on instruction tuned model. Note that tuning base models can be regarded as live-line work, meaning that we hope to enhance models' safety without sacrificing models' general instruction following capabilities in other aspects. We find that `SN-Tune` effectively enhances model safety without compromising general capabilities, and in some cases, even slightly improves them. Specifically, `SN-Tune` reduces the harmful score of Vicuna-13B-v1.5 from 93.5 to 3.0, Llama3-8B-Instruction from 65.5 to 2.0, and Mistral-7B-Instruct-v0.2 from 70.8 to 4.5. Meanwhile, the general capabilities are largely preserved. Furthermore, compared to Circ-Break, `SN-Tune` requires less training time and fewer additional parameters.

## 3.2 EFFICIENT ESTABLISH SAFE MECHANISM FOR BASE MODEL

**Experimental Settings** When implementing `SN-Tune` on base models, we largely maintain the settings described in Section 3.1, with two key differences. First, we do use the specific chat template for fine-tuning. Second, for evaluations on GSM8K, we employ a 5-shot approach rather than zero-shot, given the use of base models.

**Main Results** Table 3 shows the performance of `SN-Tune` on base models. We find that `SN-Tune` effectively enhances model safety without compromising general capabilities, and in some cases, even slightly improves them. Specifically, `SN-Tune` reduces the harmful score of Llama2-7B-Base from 98.8 to 5.3, Llama3-8B-Base from 100.0 to 13.5, and Mistral-7B-v0.1 from 100.0 to 13.8. Meanwhile, the general capabilities are largely preserved. For instance, the original general capability score for LLama2-7B-Base is 29.8, while the model after `SN-Tune` achieves 31.6. Similarly, the score increases from 66.6 to 67.0 for Llama3-8B-Base and from 100.0 to 13.8 for Mistral-7B-v0.1. Furthermore, different from instruction-tuned models, Circ-Break can not construct safety mechanism on the base model with several training corpus. Specifically, harmful score of Llama2-7B-Base after tuned by Circ-Break is still 90.5, while the number is 92.5 for Llama3-8B-Base and 87.5 for Mistral-7B-v0.1. Moreover, the training time for `SN-Tune` on Llama2-7B-Base is just 2 minutes, while Circ-Break requires 23 minutes. On Llama3-8B-Base, the time costs are 2 and 35 minutes respectively, while on Mistral-7B-v0.1, they are 2 and 21 minutes respectively.

Table 4: `RSN-Tune`'s performance on improving models' safety robustness. "Before": pre-tuning. "Original": direct tuning. "SN-Tune" and "RSN-Tune": tuning on safety-enhanced models.

| Dataset | Llama2-7B-Chat | | | | Mistral-7B-Instruct-v0.2 | | | |
|---|---|---|---|---|---|---|---|---|
| | Before | Origin. | `SN-Tune` | `RSN-Tune` | Before | Origin. | `SN-Tune` | `RSN-Tune` |
| **GSM8K** | 16.8 | 26.5 | **27.2** | 26.2 | 43.7 | **63.4** | 61.8 | 63.2 |
| **Harmful** | 0.0 | 41.0 | 38.0 | **26.0** | 36.0 | 79.0 | 72.0 | **41.0** |

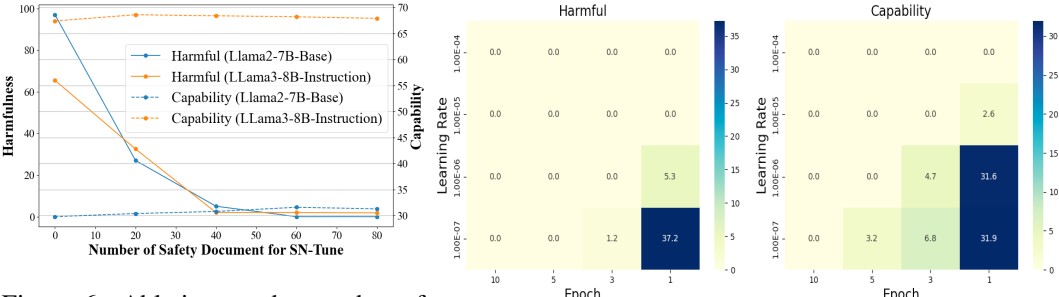

Figure 6: Ablation on the number of safety documents used in training.

Figure 7: Ablation on training epoch and learning rate..

## 4 MORE ROBUST EFFICIENT SAFETY TUNING

Fine-tuning instruction-tuned models on specific downstream tasks is a common practice for users seeking to optimize performance in particular application scenarios (Yu et al., 2024; Zhao et al., 2024c). However, Qi et al. (2024); Jain et al. (2024) have noted that even fine-tuning with seemingly benign and widely used datasets can unintentionally compromise the safety alignment of LLMs. To address this issue and mitigate its effects, we propose a Robust Safety Neuron Tuning method, called `RSN-Tune`. According to Zhao et al. (2024b), a specialized set of neurons, termed foundation neurons, are responsible for fundamentally managing queries. Consequently, the vulnerability of a model's safety mechanism to general fine-tuning may be attributed to the overlap between foundation neurons and safety neurons, with the latter being inadvertently altered during the fine-tuning process. Inspired by this observation, we propose separating the safety neurons from the foundation neurons. This separation is achieved by selectively tuning only those safety neurons that do not overlap with foundation neurons when applying `SN-Tune` to instruction-tuned models, as illustrated in Section 3.1. We then conduct experiments to prove the effectiveness of `RSN-Tune`.

**Experiment Settings**  We employ Llama2-7B-Chat and Mistral-7B-Instruct-v0.2 as backbone models considering their excellent safety performance and generality. For fine-tuning, we employ the GSM8K dataset (Cobbe et al., 2021), widely recognized as a challenging and representative benchmark for reasoning tasks. The foundation neurons are detected by Wikipedia corpus[2] with the same neuron detection method illustrated in Section 2.1.

**Main Results**  Table 4 demonstrates the effectiveness of `RSN-Tune` in enhancing models' safety robustness during downstream tuning. We observe that direct tuning using the GSM8K training set significantly increases model harmfulness. For instance, Llama2-7B-Chat's harmful score rises from 0.0 to 41.0, while Mistral-7B-Instruct-v0.2's score increases from 36.0 to 79.0. This phenomenon also affects `SN-Tune`, which indiscriminately enhances all safety neurons, regardless of their overlap with foundation neurons. In contrast, `RSN-Tune` partially preserves model safety after downstream tuning. Specifically, it reduces Llama2-7B-Chat's harmful score to 26.0 and Mistral-7B-Instruct-v0.2's to 41.0. However, a complete harmful score reduction to 0.0 is not achievable due to an insufficient number of non-overlapping safety neurons.

## 5 Further Analysis

In this section, to further understand the mechanism and explore the influencing factors to the performance of `SN-Tune`, we conduct comprehensive ablation analysis, mainly including the number of training safety documents, training epoch and learning rate.

### 5.1 Number of Safety Documents for `SN-Tune`

**Experiment Settings**   We employ LLama2-7B-Base to serve as the representative base model and Llama3-8B-Instruction to represent the instruction-tuned model. Following the setting outlined in Section 3, we assess the models' overall performance and potential harmfulness after tuning by `SN-Tune` with varying quantities of safety-related documents.

**Main Results**   Figure 6 illustrates the effect of training document quantity on `SN-Tune`. We observe that the general capabilities of both LLama2-7B-Base (yellow dotted line) and Llama3-8B-Instruction (blue dotted line) remain largely unaffected regardless of the training document size. This stability is primarily attributed to the limited number of neurons trained. Specifically, as we only train the safety neurons, which comprise approximately $0.5\%$ of all parameters, the majority of the language ability remains intact, resulting in preserved general capabilities. Notably, the harmful score of both models decreases rapidly as the number of training documents increases to 40 for LLama2-7B-Base (yellow line) and Llama3-8B-Instruction (blue line). This demonstrates the efficiency of `SN-Tune` in both enhancing and establishing model safety mechanism with just a few dozen documents. In contrast, Circ-Break requires around 4000 safety documents and a retention dataset of similar size (Zou et al., 2024). These findings underscore that `SN-Tune` is not only effective but also highly efficient in tuning safety for LLMs.

### 5.2 Learning Rate & Training Epoch

**Experiment Settings**   We further explore the effects of learning rate and number of training epochs simultaneously, as both hyperparameters influence the magnitude of parameter updates. We employ Llama2-7B-Base as our model since instruction-tuned versions derived from it are highly representative of safe language models. Similar to Section 5.1, we investigate the model's performance in terms of both general capabilities and safety aspects.

**Main Results**   Figure 7 illustrates the impact of learning rate and training epoch on both harmfulness (left) and general capability (right). We observe that with 10 training epochs, harmful score reaches 0.0, but the model also loses generality, scoring 0.0 in capability. As the number of epochs decreases, this effect diminishes. For instance, with 5 epochs and a learning rate of $10^{-7}$, the general capability improves to 3.2. Further reducing to 3 epochs maintains low harmful scores across all learning rates while increasing general capability to 6.8 at a $10^{-7}$ learning rate. The best performance is achieved with a single epoch, aligning with other continue-train approaches (Dou et al., 2024; Zhang et al., 2024). Additionally, higher learning rates lead to overfitting, resulting in both harmful score and general capabilities dropping to 0.0, while lower rates fail to effectively train safety into the model. Consequently, a learning rate of $10^{-6}$ emerges as the optimal balance between low harmful score and high general capability.

## 6 Related Work

**Safety Alignment.**   To build safe LLMs, alignments has also been a widely studied topic in the community (Stiennon et al., 2020; Ouyang et al., 2022). Efforts have been put into improving helpfulness (Bai et al., 2022; Cheng et al., 2023), honesty (Kaddour et al., 2023; Liu et al., 2023; Park et al., 2023), and harmlessness (Hartvigsen et al., 2022). Among them, safety, i.e., reducing harmfulness, is established and improved via optimization (Ouyang et al., 2022; Rafailov et al., 2024; Yuan et al., 2023), refining training data (Zhou et al., 2024; Rafailov et al., 2024; Zhang et al., 2024), or implementing additional structures designed to intentionally block harmful outputs (Inan et al., 2023; Zou et al., 2024). However, these methods are indirect and require many resources.

---

[2]`https://huggingface.co/datasets/wikimedia/wikipedia`

**Interpretability.** In the era of LLMs, one brunch of interpretability work includes efforts to understand knowledge storage (Geva et al., 2021; Dai et al., 2022; Geva et al., 2022; Meng et al., 2022; Li et al., 2023; Kotha et al., 2024; Jain et al., 2024). Another line of research centers on the self-attention layer, examining its connection to reasoning capability (Hou et al., 2023; Stolfo et al., 2023; Friedman et al., 2023) by contrasting the reasoning tree based on attention weights. In the context of safety, prior works tried to identify and interpret safety mechanisms in LLMs from either layer-level (Li et al., 2024) or feature-level (Chen et al., 2024). However, their identification methods attribute nearly 10% of parameters to safety-related functions, which is too coarse to be used.

**Interpret Safety Mechanism.** Some workers try to interpret the safety mechanism of LLMs. Wei et al. (2024b) identifies safety neurons using the SNIP score (Lee et al., 2019), which requires correct labels and enforces sparsity constraints (Sun et al., 2024). In contrast, our method operates without correct labels or uniform sparsity, identifying only 0.1% of parameters as safety neurons (compared to 3% in (Wei et al., 2024b)) with higher accuracy. Additionally, while Wei et al. (2024b) focuses on analyzing safety neurons, we introduce SN-Tune and RSN-Tune to enhance LLM safety alignment, which Wei et al. (2024b) does not address. Other works, such as Chen et al. (2024), Hsu et al. (2025), and Zhang et al. (2022), take different approaches to model safety. Chen et al. (2024) limits detection to the feed-forward layer, identifying 5% of parameters as safety neurons. Hsu et al. (2025) incorporates structural modifications, while Zhang et al. (2022) focuses on attack improvements, an impractical approach for black-box LLM training.

## 7 CONCLUSION

Safety alignment in LLMs is critical yet underexplored. We introduced a method to detect and tune safety neurons, which are less than 1% of parameters and mainly in self-attention layers. Our Safety Neuron Tuning (`SN-Tune`) enhances model safety without compromising performance, significantly reducing harmful scores in both instruction-tuned and base models. This approach also improves safety robustness during fine-tuning by separating safety neurons from foundational ones.

## ACKNOWLEDGMENTS

This research is partially supported by the National Research Foundation Singapore under the AI Singapore Programme (AISG Award No: AISG2-TC-2023-010-SGIL) and the Singapore Ministry of Education Academic Research Fund Tier 1 (Award No: T1 251RES2207). We thank Shiqi Chen for the insightful discussion at the beginning of the project.

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

## A  APPENDIX

### A.1  PARALLEL NEURON DETECTION METHOD

**Feed-Forward Network (FFN)**   In the latest open-source models, when processing input $c$, the feed-forward network in a certain layer is defined as

$$\text{FFN}(x) = \Big(\text{SiLU}\big(W_{gate}(x)\big) \cdot W_{up}(x)\Big)W_{down}, \tag{7}$$

where $x \in \mathbb{R}^{l \times d_{model}}$ is the embedding fed into the FFN, $W_{gate}, W_{up} \in \mathbb{R}^{d_{model} \times d_{inter}}$[3], $W_{down} \in \mathbb{R}^{d_{inter} \times d_{model}}$. The calculation of the importance of the $k$-th neuron in $W_{up}$, when processing the input $c$, as presented in Equation 2, can be equivalently transformed to

$$\text{Imp}(W_{up}[:,k]|c) = \|\hat{\text{FFN}}(x) - \text{FFN}(x)\|_2 = \left\|\big(h_{\text{ffn}}(x) \cdot \text{Mask}[k]\big)W_{down}\right\|_2, \tag{8}$$

where $h_{\text{ffn}} \in \mathbb{R}^{l \times d_{inter}}$ represents the embedding before $W_{down}$, and $\text{Mask}[k] \in d_{inter}$ is a vector with the $k$-th element equal to 1 and the rest equal to 0. To calculate $\text{Imp}(W_{up}[:,k]|c)$ for $k \in d_{inter}$ parallelly, we introduce a diagonal mask matrix of size $(d_{inter}, d_{inter})$, denoted as $\text{Mask}$. Therefore,

$$\text{Imp}(W_{up}|c) = \|(h_{\text{ffn}}(x) \cdot \text{Mask})W_{down}\|_2. \tag{9}$$

Furthermore, we observe that deactivating the $k$-th neuron of $W_{down}$ is equivalent to deactivating the $k$-th neuron in $W_{up}$, as they both result in $h_{\text{ffn}}[k] = 0$. Hence, we can also derive $\text{Imp}(W_{down}|c)$ by employing Equation (9).

---

[3]$W(\cdot)$ represents the linear matrix product of the input $x$ and the parameter $W$, i.e., $W(x) := xW$.

**Self-Attention Network** When processing input $c$, the self-attention network in a certain layer is

$$\text{Attention}(x) = \text{Softmax}\left(\frac{W_Q(x)W_K^T(x)}{\sqrt{d}}\right)W_V(x), \tag{10}$$

where $W_Q, W_K, W_V \in \mathbb{R}^{d_{model} \times d_{mid}}$. [4] Since $W_V(x)$ is not in the non-linear softmax calculation, we can calculate $\text{Imp}(W_V|c)$ by applying Equation (9). For $W_Q$, we obtain $\text{Imp}(W_Q[:,k]|c)$ by deactivating its $k$-th neuron, specifically, $\hat{W}_Q \leftarrow W_Q[:,k] = 0$. Firstly, we calculate the difference in attention weight before and after deactivation, prior to scaling and softmax,

$$\Delta_k(x) = \hat{W}_Q(x)W_K^T(x) - W_Q(x)W_K^T(x) = W_Q(x)[:,k]W_K(x)[k,:] \in \mathbb{R}^{l \times l}. \tag{11}$$

Next, as the changes in attention exhibit a positive correlation with the changes in the output of this layer, the importance of $W_Q[:,k]$ in processing $c$, can be approximated as

$$\begin{aligned}
\text{Imp}(W_Q[:,k]|c) &\approx \|\hat{\text{attention}}(x) - \text{attention}(x)\|_2 \\
&\approx \left\|\text{softmax}\left(\frac{W_Q(x)W_K^T(x) - \Delta_k(x)}{\sqrt{d}}\right) - \text{softmax}\left(\frac{W_Q(x)W_K^T(x)}{\sqrt{d}}\right)\right\|_2.
\end{aligned} \tag{12}$$

This process can also be calculated in parallel, specifically,

$$\begin{aligned}
\Delta(x) &= \hat{W}_Q(x)W_K^T(x) - W_Q(x)W_K^T(x) \\
&= W_Q(x).resize(l, 1, d_{mid}) \times W_K(x).resize(1, l, d_{mid}) \in \mathbb{R}^{l \times l \times d_{mid}}.
\end{aligned} \tag{13}$$

Therefore, the importance of $W_Q$ in processing input $c$ is calculated by

$$\text{Imp}(W_Q|c) \approx \left\|\text{softmax}\left(\frac{W_Q(x)W_K^T(x) - \Delta(x)}{\sqrt{d}}\right) - \text{softmax}\left(\frac{W_Q(x)W_K^T(x)}{\sqrt{d}}\right)\right\|_2. \tag{14}$$

Similarly, since $W_K$ is symmetrical to $W_Q$, $\text{Imp}(W_K|c)$ can be calculated in the same way.

## A.2 SAFETY NEURON DETECTION CORPUS

In the neuron detection process, we utilize the training documents from Zou et al. (2024), from which sampling 200 documents for detection. Specifically, the training set contains harmful queries across various categories, including "terrorism and violent extremism", "self-harm", and "political campaigning", etc. This diverse dataset helps ensure the generalizability of the detected neurons. Furthermore, our analysis examined how the number of input documents affects safety neuron detection, as shown in Table 5. The ablation analysis is on Llama3-8B-Instruct, and the results demonstrate that 200 documents are sufficient to reliably identify safety neurons.

Table 5: Number of detected safety neurons across different document sizes.

| Corpus Size | 10 | 50 | 100 | 200 | 400 | 800 |
|---|---|---|---|---|---|---|
| **Number of Safety Neurons** | 8912 | 4825 | 3594 | 2329 | 2322 | 2314 |

---

[4]In some models like Vicuna and Mistral, $d_{model} = d_{mid}$, but we use different notations to avoid ambiguity.

