# OpenReview forum: "Understanding and Enhancing Safety Mechanisms of LLMs via Safety-Specific Neuron"
_ICLR.cc/2025/Conference — ICLR 2025 Poster_

### Official Review · Reviewer_WgZu · 2024-10-26

**Soundness:** 2
**Presentation:** 3
**Contribution:** 2
**Rating:** 3
**Confidence:** 5

**Summary:**

This paper identifies safety neurons that are associated with safety alignment in LLMs. The authors claim that the safety neurons constitute less than 1% of all parameters, and they are predominantly located in self-attention layers. Experimental results have been provided

**Strengths:**

This paper deals with an important topic, the safety issue of LLMs – for safety alignment.

**Weaknesses:**

- My biggest concern is the distinction between paper, Zhao et al (2024b) (How do Large Language Models Handle Multilingualism? Yiran Zhao, Wenxuan Zhang, Guizhen Chen, Kenji Kawaguchi, Lidong Bing 2024).  I believe this submission’s section 2.1 is the meat of the work, but the formulations are all found in Zhao et al (2024b) – just the same. Then, this submission’s distinctive contribution is very unclear. The only difference is (3) where the authors feed harmful queries and see the activations (although the sentence in lines 124-125 is broken, so it is speculated. It is impossible to know their actual implementation.) However, the entire formulations is just the same with the other paper, with only a minor tweak with harmful queries for safety context.

- Moreover, the authors failed to properly provide a reference for the formulation/equation (1) and (2) and give credit to the authors of the paper. The paper is referred to only for the parallel neuron detection method. ((4), (5), and (6)).

-	Moreover, in terms of the paper’s direction and theme, this submission’s distinctive novelty from the following paper is unclear: “Assessing the Brittleness of Safety Alignment via Pruning and Low-Rank Modifications,” Boyi Wei · Kaixuan Huang · Yangsibo Huang · Tinghao Xie · Xiangyu Qi · Mengzhou Xia · Prateek Mittal · Mengdi Wang · Peter Henderson, ICML 2024. This paper also identifies safety components in LLMs.

-	Unlike the authors’ statement, “Regarding general capability, deactivating the safety neuron shows minimal impact, “similar to deactivating randomly selected neurons”,……” in Table 1, Deact_SN’s Avg. Capability is constantly, always lower than Deact-R, which hints that the safety neurons are also contributing to general capability. If the weights were purely safety neurons, when they were pruned, they shouldn’t impact the general capability even with no need to compare it with Deact-R. But, the table result shows there exists an impact on general capability.

-	In order to conclude “Safety neurons predominantly reside within the self-attention layers.”, the authors should have reported the original number of neurons and the proportion of safety neurons in feed-forward and self-attention layers, respectively – not just the split.

-	For section 3, “Efficient Safety Training” the authors claimed SN-Tune is efficient by comparing the training cost with Circ-Break. However, SN-Tune requires a process and resources and effort to “identify” the safety neurons. Only after the neurons are obtained, the SN-Tune can be applied. Hence, it is not a fair comparison. The cost/effort for identifying the safety neurons must be taken into account, but they were not in this submission.

-	It is unclear the impact of RSN-Tune compared to SN-Tune by looking at the results of GSM8K in Table 4.

-	This submission lacked an important and nominal paper in reference, with regard to neuron importance, Pavlo Molchanov, Arun Mallya, Stephen Tyree, Iuri Frosio, and Jan Kautz. Importance estimation for neural network pruning. 2019 IEEE/CVF Conference on Computer Vision and Pattern Recognition (CVPR).

**Questions:**

* From which figure/table can we infer that “1%” are safety neurons?
* Why were helpfulness scores not reported?
* Why were Llama2 instruction-tuned models not reported in Table 2? It is known that Llama2 is more resilient than Llama3 in terms of instruction following. Hence, the Llama2's results for Fig 2 need to be reported.
* What’s language X and Y in Fig 5?
* Where are the results of the five languages?
* What is the objective of having section 2.3.2 regarding the theme of the paper?
* What is “Language-specific neurons” in line 291? It has not been defined nor introduced beforehand.

---

> ### Author Response · Authors · 2024-11-20
>
> Dear Reviewer WgZu,
>
> We appreciate the time and effort you have put into providing valuable feedback. However, we respectfully believe there might be some misunderstanding regarding our work. We would appreciate the opportunity to clarify a few points and address your concerns as follows:
>
> >  Concern #1 Distinction of the paper
>
> Our safety neuron detection method builds upon the approach introduced in [1], but our work significantly extends this foundation. Instead of proposing a new detection method, our key contribution is applying this technique to safety scenarios. We define, identify, and verify safety neurons while conducting a comprehensive analysis to explore their features. Furthermore, based on our novel and insightful findings, we develop two novel methods: SN-Tune for efficient safety alignment and RSN-Tune for robust safety alignment.
>
> Another paper you mentioned [2] similarly applies existing techniques to safety scenarios rather than introducing new detection methods. It relies on SNIP scores [3] which require ground truth labels, and implements [4]'s uniform sparsity constraints. In contrast, our method achieves superior performance without these limitations - identifying less than 0.3% of parameters as safety neurons versus their 3%, while maintaining higher accuracy. Moreover, we extend beyond neuron analysis to introduce practical safety alignment methods (SN-Tune and RSN-Tune), an aspect not addressed in [2].
>
> We respectfully note that multiple reviewers have recognized our contributions: Reviewer 9nQR noted our "interesting verification of safety neurons," Reviewer Uh2w highlighted our "originality," "experimental rigor," and "practical impact," Reviewer Qmyn praised our "strong defense performance," and Reviewer R3v2 emphasized our "empirical insights" and "relevance of results for practitioners."
>
> > Concern #2 Features of safety neurons
>
> 1. Safety neurons are also used for general capability.
>
> We would like to first clarify “pruning” that you mentioned in the review and “deactivating” that we used in our experimental settings. In the context of neuron level, pruning cannot be directly applied to the self-attention layer, as it would alter the number of key-value pairs in each attention head. This limitation can only be overcome by either enforcing uniform neuron removal across heads or eliminating entire attention heads. In contrast, deactivation can be implemented in any structure by simply setting the parameters of the neuron to zero. While the sparsity approach employed in [2] is applicable to self-attention structures, it proves inefficient due to its high computational cost and low accuracy, as it requires maintaining uniform sparsity across columns or rows. This flexibility enables our method for more accurate safety neuron detection.
>
> Moreover, we acknowledge that completely separating safety-related neurons from general capability neurons is challenging, as safety itself represents a specific model capability. This observation aligns with [2], where the phenomenon is even more severe. As shown in their Figure 2(a), the safety mechanism completely breaks down when general capability accuracy decreases from 0.58 to 0.52. Additionally, while [2] evaluates general capabilities using mostly classification tasks, our evaluation involves the more challenging GSM8K reasoning tasks.
>
> 2. Safety neurons in different structures.
>
> Since self-attention layers contain approximately half as many neurons as feed-forward layers, the higher concentration of safety neurons in self-attention shown in Figure 3 indicates that safety neurons are disproportionately located in self-attention layers. To further clarify and facilitate understanding, we list the number of safety neurons in each architectural component.
>
> |                          | Llama3-8b-Instruction | Llama2-7b-chat | Mistral-7b-instruct-v0.2 |
> | :----------------------- | :------------: | :------------: | :----------------------: |
> | Number in Self-Attention |      1787      |      3801      |           1037           |
> | Number in Feedforward    |      1142      |      1135      |           608            |
>
> >  Concern #3 Experiment details for SN_Tune and RSN-Tune.
>
> 1. Safety Neuron detection time
>
> Thanks for the comment. We acknowledge this oversight in the paper. For clarification, we utilize the training documents from [5] as noted in line 158, and sample 200 documents for detection. Due to the parallel detection approach described in Equations (4) and (5), the entire detection process is completed in under one minute.
>
> 2. Difference between SN-Tune and RSN-Tune
>
> According to Table 4, the RSN-Tuned model demonstrates reduced harmfulness compared to the SN-Tune, with scores decreasing from 38.0 to 26.0 for Llama2-7b-chat and from 72.0 to 41.0 for Mistral-7B-Instruct-v0.2. As noted in lines 412-413, "A complete harmful score reduction to 0.0 is not achievable due to an insufficient number of non-overlapping safety neurons."

---

> ### Author Response · Authors · 2024-11-20
>
> >  Concern #4 Unclear explanation
>
> We clarify several details from the paper that may have caused confusion. We will update the draft to make it clearer.
>
> 1. We explain the portion of safety neurons from line 245 to line 247. “In Llama2-7B-Chat, the leap appears when deactivating 0.3% neurons, while the number is 0.15% for Llama3-8B-Instruction and is 0.1% for Mistral-7B-Instruct-v0.2.”
> 2. Thanks for your valuable suggestion. However, helpfulness is not explored in this paper and we mainly focus on harmfulness and general capability, a setting same as [5]. Similarly, LLama2-7b-chat is not reported in Table 2 because its harmfulness has already been very low and leaves no space for us to improve, also the same setting as [5].
> 3. Section 2.3.2 analyzes safety neurons from a multilingual perspective, as safety alignment across different languages poses significant challenges. In Figure 5, languages X and Y represent the language pairs for which we analyze safety neuron overlap, following the notation used in Equation (288).
> 4. Thanks for reminder. We ignore detailed description of language-specific neurons in the paper, which are a subset of neurons responsible for processing multilingual queries, as defined in [1].
>
>
>
> [1] Zhao, Yiran, et al. "How do Large Language Models Handle Multilingualism?." arXiv preprint arXiv:2402.18815 (2024).
>
> [2] Wei, Boyi, et al. "Assessing the brittleness of safety alignment via pruning and low-rank modifications." arXiv preprint arXiv:2402.05162 (2024).
>
> [3] Lee, Namhoon, Thalaiyasingam Ajanthan, and Philip HS Torr. "Snip: Single-shot network pruning based on connection sensitivity." International Conference on Learning Representations (2022).
>
> [4] Sun, Mingjie, et al. "A Simple and Effective Pruning Approach for Large Language Models." The Twelfth International Conference on Learning Representations. (2024)
>
> [5] Zou, Andy, et al. "Improving alignment and robustness with circuit breakers." The Thirty-eighth Annual Conference on Neural Information Processing Systems. (2024)

---

> ### Author Response · Authors · 2024-11-23
>
> Dear Reviewer WgZu,
>
> Thank you for your thoughtful comments and valuable feedback. We deeply appreciate the time and effort you have invested in reviewing our work.
>
> We have carefully addressed your comments in the rebuttal, clarifying key details to respond to your concerns, suggestions, and misunderstandings. We kindly ask for your prompt attention to our response.
>
> Thank you again for your time and consideration.
>
> Best regards, Authors

---

> ### Comment · Reviewer_WgZu · 2024-11-30
>
> Thank the authors for their response. However, there are concerns as below.
>
> >	Concern #2 Features of safety neurons
>
> **1.	Safety neurons are also used for general capability.**
>
> •	What you referred to as “deactivating” is, in fact, magnitude (a.k.a. weight or unstructured) pruning, which I referred to.
>
> • As for the fact that this submission still has a huge overlap with [1], the response was not convincing. All the exact same formulations are found in [1] except for (3), where the authors feed harmful queries and see the activations. In addition, [2] also identified safety weights. Hence, considering those closely related papers, the contribution and novelty of this submission are not significant.
>
> •	The authors’ claim in the paper “Regarding general capability, deactivating the safety neuron shows minimal impact” conflicts with the claim in the response, “completely separating safety-related neurons from general capability neurons is challenging” Also, I noticed that although the authors stated them as “safety neurons” in the paper, in the response they slightly change the wording as “safety-related neurons.”
>
> \
> **2.	Safety neurons in different structures.**
>
> •	The table confirms that the authors’ statement was overclaimed, “Safety neurons predominantly reside within the self-attention layers.”. 8:5 or 5:3 ratios are hard to say as “predominant.”
>
>
> >Concern #4 Unclear explanation
>
> I note that it is not fair that the authors selectively do not show helpfulness and results on Llama2 instruction-tuned models, especially for fair and confident comparisons with other safety studies in LLMs. Hence, I am not fully convinced by this work.
>
> \
> By considering all the points above, I maintain the original score.
>
> [1] Zhao, Yiran, et al. "How do Large Language Models Handle Multilingualism?." arXiv preprint arXiv:2402.18815 (2024).
>
> [2] Wei, Boyi, et al. "Assessing the Brittleness of Safety Alignment via Pruning and Low-Rank Modifications." ICML (2024).

---

> ### Author Response · Authors · 2024-12-03
>
> Thank you Reviewer WgZu for your detailed reply! Let us address your remaining concerns as follows:
>
> > Llama2 instruction-tuned models
>
> Like we mentioned earlier, we followed our baseline [1] and did not report results on Llama2-7B-Chat results in table 2 because it is already safe and resistant to safety attack methods.
>
> However, we conducted experiments to use SN-Tune to make Llama2-7b-base safe, which can be found in table 3. The results show that our method can effectively make the base model even safer than the released chat model meticulously trained to ensure its safety by a big company.
>
> |              | Llama2-7b-base | Llama2-7b-base + SN-Tune | Llama2-7b-chat |
> | ------------ | :------------: | :----------------------: | :------------: |
> | Direct       |      97.0      |         **0.0**          |    **0.0**     |
> | GCG          |     100.0      |         **7.0**          |      31.0      |
> | AutoDAN      |     100.0      |           9.0            |    **0.0**     |
> | PAIR         |      98.0      |         **5.0**          |      6.0       |
> | Avg. Harmful |      98.8      |         **5.3**          |      9.3       |
>
> We will include this result in the paper.
>
>
>
> > Contribution of the paper
>
> We have elaborated the contribution of the paper in our previous response. Having [2] as our prior work by no means diminishes our contribution. Our paper and [2] studies two completely different fields: multilinguality and safety. None of our proposed methods, experiments and findings appear in [2].
>
> The other reviewers gave 8, 6, 8, 6 for a reason. We identify individual neurons and train them and it’s different from the methods in prior work [3] that you mentioned. First, the biggest difference in method is that they study pruning methods to identify relevant parameters while we identify neurons and train them. They only show the impact of pruning those parameters instead of trying to make the model safe without sacrificing the capacity. As a result, our method as shown in table 2 can achieve safety with equal or better capability in reasoning tasks, while their studied methods lead to significant drop in classification accuracy when they prune those parameters as shown in the Figure 2(a) of their paper. Lastly, we only identify 0.3%, 0.15% and 0.1% of the neurons for various models to be safety neurons or safety-related neurons while they identify 3% of the parameters. We hope that this clarifies your question about the comparison with [3].
>
> To us, being able to train individual neurons is like performing surgery on neuron networks. We do not know how general this method is but it is at least interesting and may have the potential to be used in other domains.
>
>
> > 8:5 or 5:3 ratios are hard to say as “predominant.”
>
> We searched the meaning of predominant in legal documents. According to this database [4], predominant means more than 80% or 50%. The average being 65% roughly matches 8:5 and 5:3 ratios. We are happy to change this word to “largely” if that is a better word. But we certainly did not say “almost all”, which would be an inaccurate description. Thank you for pointing this out!
>
> > general capability
>
> As shown in Table 1, Deact-SN is 1 or 2 points lower than Deact-R in Avg. Capability when the Avg. Capability scores are 34.8, 67.6 and 50.3 which implies that "deactivating the safety neuron shows minimal impact" and “completely separating safety-related neurons from general capability neurons is challenging” are both true at the same time. We think it is fine to say safety-related neurons here.
>
> > “deactivating” is, in fact, magnitude (a.k.a. weight or unstructured) pruning
>
> We’ve provided the definition of deactivating in line 708 and line 698 of the paper. We do not think there is any ambiguity here. Magnitude pruning is a specific terminology in the pruning field which is not familiar to the safety community and general LLM community, in addition to the other implications we discussed in the previous reply. Thus we will use our original terminology “deactivating”.
>
> > score
>
> It is interesting that the reviewer does not mention any strengths of our work except that we work on safety, which is an important topic. This definitely is a completely different view than our own views and other reviewers’ views. We have done our best to address the reviewer’s comments but we are honestly not sure if the reviewer is willing to change their opinion.
>
>
>
>
>
> [1] Zou, Andy, et al. "Improving alignment and robustness with circuit breakers." NeurIPS (2024).
>
> [2] Zhao, Yiran, et al. "How do Large Language Models Handle Multilingualism?." NeurIPS (2024).
>
> [3] Wei, Boyi, et al. "Assessing the brittleness of safety alignment via pruning and low-rank modifications." ICML (2024).
>
> [4] https://www.lawinsider.com/dictionary/predominantly

---

### Official Review · Reviewer_R3v2 · 2024-11-02

**Soundness:** 3
**Presentation:** 3
**Contribution:** 3
**Rating:** 8
**Confidence:** 4

**Summary:**

The authors first introduce a technique for detecting LLM neurons (i.e. rows or columns in LLM weight matrices) responsible for safety (e.g. refusing to comply with harmful instructions). The method is fairly simple: a neuron is said to be a safety neuron if deactivating it changes the final next-token embedding by at least some threshold (measured in Euclidean distance) on each of a set of inputs. They validate that deactivating these neurons indeed increases harmfulness scores of three open-weight models, while preserving performance on capabilities benchmarks.

They then propose a safety fine-tuning method, named SN-Tune, which tunes only these safety neurons, keeping the remaining ones fixed. They show SN-tune significantly reduces harmfulness scores across several open weight models; e.g. Llama3-8B’s score decreases from 65.5 to 2.0. Finally, they propose a refinement of SN-Tune, termed RSN-Tune, which also identifies neurons responsible for model capabilities, and only tunes safety neurons which do not overlap with such capabilities neurons, hence mitigating capability degradation during safety tuning. Additional experiments include a study of the overlap of safety neurons across languages (which they find to be small) and of applying SN-tune to pre-trained models (rather than already-fine-tuned models).

**Strengths:**

1. Simplicity of safety neuron detection: their notion of safety neuron is conceptually simple and can be computationally ascertained (instead of relying on manual analyses). Their later experiments indicate that one can indeed obtain a useful safety tuning algorithm by using their safety neuron detection method.

1. Empirical insights into how safety mechanisms are implemented in LLMs: the paper’s results indicate that safety neurons are very concentrated (1% of parameters), occur primarily in self-attention modules and in earlier layers, and are mostly not shared across languages. These insights seem relevant for the community as a whole, particularly when devising defenses to multi-lingual attacks.

1. Relevance of results for practitioners: one of the biggest hurdles in safety tuning is the performance degradation it usually entails if done in a naive way. The results in the paper suggest SN-tune might be useful for practitioners looking to safety-tune models without harming their capabilities. In particular, one could in principle automatically detect safety neurons using a small corpus of examples, rather than requiring manual analyses or large corpora.

1. The paper is written mostly clearly and was easy to follow.

**Weaknesses:**

1. Lack of detail on how foundation neurons are detected: in Section 4, the authors propose not tuning safety neurons that also serve as foundation neurons (i.e. play  a role in model capabilities). However, they do not clarify how they detect the foundation neurons. This harms the reproducibility of the RSN-tune results.

1. Notation and presentation in Section 2.1: I would recommend revising the presentation in the Accelerated Safety Neuron Detection. For example, the authors write that Mask is an identity matrix, which, as stated, cannot be true. Looking at the appendix, it seems they might mean it is a binary matrix (i.e. the entries are 0 or 1). Also, the authors could clarify what they mean by parallel computations. Does this simply mean adding many computations in a batch? Generally, it seems that their notion of safety neuron is embarrassingly parallelizable, in the sense that their criterion can be computed independently for all neurons.

1. Missing citation of Kotha et al. (2023) in section 2.3.2: one relevant work concerning multi-lingual attacks on LLMs is Understanding Catastrophic Forgetting in Language Models via Implicit Inference, by Kotha et al. 2023. They show that translating harmful instructions into low-resource languages can elicit harmful responses, which can then be translated back to the original language. Their results are complementary to the author’s findings that safety neurons have little overlap across languages.

1. Another work the authors do not discuss, but which I believe is relevant and complementary to their results, is Mechanistically analyzing the effects of fine-tuning on procedurally defined tasks, by Jain et al. (2024). Amongst other results, Jain et al. also find that the effects of fine-tuning can be neutralized by pruning a small number of neurons.

**Questions:**

1. Would it be correct to conclude from lines 362-363 that your results indicate that base models already contain safety neurons, despite not being safety-tuned?

1. It could be useful to have a study on the distribution of safety neurons in the base models to verify they share similar properties to those in fine-tuned models, i.e. being concentrated (<1% of parameters) and being predominantly present in earlier layers and attention modules.

1. Similarly, one could study whether their harmfulness scores get somehow even worse when these neurons are disabled. To make this non-trivial, you could have “safe-behavior-inducing” few-shot examples in the prompt, for example.

---

> ### Author Response · Authors · 2024-11-20
>
> Dear Reviewer R3v2,
>
> Thank you for your insightful reviews and comments. We appreciate the time and effort you have put into providing valuable feedback. We would like to address your concerns as follows:
>
> >  Concern #1 Experiment detail
>
> Thanks for the suggestion. To provide further clarity: In the safety neuron detection process, we utilize the training documents from [1] as noted in line 158, sampling 200 documents for detection. Specifically, the training set [2] contains harmful queries across various categories, including terrorism and violent extremism, self-harm, and political campaigning, etc. This diverse dataset helps ensure the generalizability of the detected neurons.
>
> Furthermore, our analysis examined how the number of input documents affects safety neuron detection. The results demonstrate that 200 documents are sufficient to reliably identify safety neurons. We will update the draft accordingly.
>
> | Llama3-8B-Chat           | 10   | 50   | 100  | 200  | 400  | 800  |
> | ------------------------ | ---- | ---- | ---- | ---- | ---- | ---- |
> | Number of Safety Neurons | 8912 | 4825 | 3594 | 2329 | 2322 | 2314 |
>
> When identifying foundation neurons, we apply the same detection algorithm but utilize a general corpus sampled from Wikipedia rather than harmful queries, as detailed in line 404. We analyze 10,000 documents, following the experimental setup established in [3]. These neurons demonstrate stable activation during fundamental language processing, validating their classification as foundation neurons.
>
>
>
> >  Concern #2 Notation and presentation in Section 2.1
>
> Thanks for the reminder. We would like to further clarify. Mask is an identity matrix with 1's on the diagonal and 0's elsewhere. For the parallel feature of neuron detection, Equations (4) and (5) operate on all neurons l in layer i simultaneously, rather than processing each neuron individually. This parallel computation across neurons enables faster execution compared to sequential processing of each neuron.
>
>
>
> >  Concern #3 Safety neurons in the base model
>
> The base model's lack of safety alignment complicates the classification of these neurons as true "safety neurons." Essentially, these neurons are activated when processing potentially harmful queries. They account for less than 1% of parameters and share a similar pattern with safety neurons in safety-aligned models. The difference is that, since the base model lacks inherent safety alignment, it consistently generates harmful outputs, regardless of the activation state of these neurons. Therefore, SN-Tune is introduced to address this by exclusively training these neurons to produce safe responses.
>
>
>
> >  Suggestion #1 More related work
>
> We appreciate your suggestion. The papers you recommend [4] [5] support our findings and serve as inspiration for our paper. We will cite them in the updated draft.
>
>
>
>
>
> [1] Zou, Andy, et al. "Improving alignment and robustness with circuit breakers." The Thirty-eighth Annual Conference on Neural Information Processing Systems. (2024)
>
> [2]https://raw.githubusercontent.com/GraySwanAI/circuit-breakers/refs/heads/main/data/circuit_breakers_train.json
>
> [3] Zhao, Yiran, et al. "How do Large Language Models Handle Multilingualism?." arXiv preprint arXiv:2402.18815 (2024).
>
> [4] Kotha, Suhas, Jacob Mitchell Springer, and Aditi Raghunathan. "Understanding Catastrophic Forgetting in Language Models via Implicit Inference." The Twelfth International Conference on Learning Representations. (2024)
>
> [5] Jain, Samyak, et al. "Mechanistically analyzing the effects of fine-tuning on procedurally defined tasks." The Twelfth International Conference on Learning Representations. (2024)

---

> > ### Comment · Reviewer_R3v2 · 2024-11-23
> > **Thank you for your response**
> >
> > I would like to thank the authors for their response, and for incorporating the feedback I included in the review. I maintain my recommendation of acceptance.

---

> > > ### Author Response · Authors · 2024-11-23
> > >
> > > Dear Reviewer R3v2,
> > >
> > > Thank you very much for reviewing our paper and reading our rebuttal. We sincerely appreciate your recognition of our contribution.
> > >
> > > We are truly grateful for your time and your reply.
> > >
> > > Authors

---

### Official Review · Reviewer_Qmyn · 2024-11-05

**Soundness:** 3
**Presentation:** 3
**Contribution:** 3
**Rating:** 6
**Confidence:** 4

**Summary:**

The paper presents a method for identifying safety neurons in Large Language Models (LLMs), which are critical for managing harmful queries and represent less than 1% of the model parameters. It introduces a technique named SN-Tune that focuses on tuning these safety neurons while maintaining the overall capabilities of the models. The findings indicate significant reductions in harmful scores, with Llama3-8B-Instruction decreasing from 65.5 to 2.0. Furthermore, the approach improves safety robustness during fine-tuning by separating safety neurons from foundation neurons.

**Strengths:**

1. The paper writing is good and very easy to follow.
2. The defense performance is strong, even compared with current SOTA defense method.
3. The logic of this article is also very coherent. The author first suggests enhancing the effectiveness of the model's safe responses through a more detailed approach to neuron control. The author also conducted relatively detailed ablation experiments to support the faithfulness of safe neurons.

**Weaknesses:**

I believe the biggest shortcoming of this paper is that the description of the experiments for identifying safe neurons is very insufficient. The author does not mention the datasets used, parameter details, or the time costs associated with the experiments. The dataset is crucial because it relates to the generalizability of the method. For example, if the author only uses datasets related to violence to identify neurons, but the identified safe neurons still demonstrate good defensive capabilities against attacks related to pornography or other safety categories, including jailbreaking attacks, it suggests that safe neurons are relatively fixed within LLMs. However, the author does not mention this point.

**Questions:**

please see weakness.

---

> ### Author Response · Authors · 2024-11-20
>
> Dear Reviewer Qmyn,
>
> Thank you for your insightful reviews and comments. We appreciate the time and effort you have put into providing valuable feedback. We would like to address your concerns as follows:
>
> >  Concern: Description of the experiments
>
> Thanks for the suggestion. To provide further clarity: In the neuron detection process, we utilize the training documents from [1] as noted in line 158, sampling 200 documents for detection. Specifically, the training set [2] contains harmful queries across various categories, including terrorism and violent extremism, self-harm, and political campaigning, etc. This diverse dataset helps ensure the generalizability of the detected neurons.
>
> Furthermore, our analysis examined how the number of input documents affects safety neuron detection. The results demonstrate that 200 documents are sufficient to reliably identify safety neurons. We will update the draft accordingly.
>
> | Llama3-8B-Instruction    | 10   | 50   | 100  | 200  | 400  | 800  |
> | ------------------------ | ---- | ---- | ---- | ---- | ---- | ---- |
> | Number of Safety Neurons | 8912 | 4825 | 3594 | 2329 | 2322 | 2314 |
>
>
>
>
>
> [1] Zou, Andy, et al. "Improving alignment and robustness with circuit breakers." The Thirty-eighth Annual Conference on Neural Information Processing Systems. (2024)
>
> [2]https://raw.githubusercontent.com/GraySwanAI/circuit-breakers/refs/heads/main/data/circuit_breakers_train.json

---

> > ### Comment · Reviewer_Qmyn · 2024-11-24
> > **Thanks for your response.**
> >
> > About "the time costs associated with the experiments.": please give some results about it.
> >
> > About your comments "the training set [2] contains ...... detected neurons.": could you show some initial results about this setting:
> > if we only use part of various categories (like only terrorism and violent extremism), could the discovered safety neurons generalize to other categories (like elf-harm, and political campaigning)?

---

> > > ### Author Response · Authors · 2024-11-24
> > >
> > > Dear reviewer Qmyn,
> > >
> > >
> > >
> > > Thank you once again for your valuable suggestion and feedback. We greatly appreciate you taking the time to provide your insights. We would like to further clarify on the points you raised.
> > >
> > >
> > >
> > > > Concern #1 Time of safety neuron detection
> > >
> > > Thank you for your comment. We acknowledge the oversight in our paper. Due to the parallel detection approach described in Equations (4) and (5), the entire detection process is completed in around one minute (43 seconds on Llama3-8b-Instruction, 34 seconds on Llama2-7b-chat, 37 seconds on Mistral-7B-Instruct-v0.2, and 62 seconds on Vicuna-13b-v1.5).
> > >
> > >
> > >
> > > > Concern #2 Detection corpus for safety neuron
> > >
> > > Thank you for bringing this up, as it is a valuable direction to explore further. We conduct additional experiments on different neuron detection corpora. Specifically, we employ three categories of harmful content, including "terrorism and violent extremism," "self-harm," and "privacy violations." We then calculate the overlap among the detected safety neurons across these different corpora. We use Llama3-8b-Instruction as the tested model. The detailed results are shown below, where the overlapping ratio is calculated in the same manner as in Figure 5 of the paper. We find that the overlapping ratios are high, around 0.9, indicating substantial overlap, but with some differences still present.
> > >
> > > |                                     | terrorism and violent extremism | self harm | privacy violation |
> > > | :---------------------------------: | :-----------------------------: | :-------: | :---------------: |
> > > | **terrorism and violent extremism** |                1                |   0.92    |       0.91        |
> > > |            **self harm**            |              0.87               |     1     |       0.89        |
> > > |        **privacy violation**        |              0.90               |   0.93    |         1         |
> > >
> > >
> > > If you have any further questions or concerns, we would be happy to engage in additional discussions. We are truly grateful for your time and comments.
> > >
> > > Best regards, The Authors

---

> > > > ### Comment · Reviewer_Qmyn · 2024-11-25
> > > > **Thanks for your response.**
> > > >
> > > > I maintain my recommendation of acceptance.

---

> > > > > ### Author Response · Authors · 2024-11-25
> > > > >
> > > > > Dear Reviewer Qmyn,
> > > > >
> > > > > Thank you very much for reviewing our paper and reading our rebuttal. We sincerely appreciate your recognition of our contribution.
> > > > >
> > > > > We are truly grateful for your time and your reply.
> > > > >
> > > > > Authors

---

### Official Review · Reviewer_Uh2w · 2024-11-07

**Soundness:** 4
**Presentation:** 4
**Contribution:** 4
**Rating:** 8
**Confidence:** 3

**Summary:**

This paper introduces the crucial concept of "safety neurons", which investigates neural activation patterns when large language models encounter unsafe instructions and jailbreak attacks. The definition of "safety neurons" is concise, clear, and effective. Furthermore, the paper presents several significant findings: the proportion of safety neurons is remarkably small, constituting less than 1% of total parameters; the robustness study of safety neuron locations reveals sensitivity decreases from front to back layers, with safety neurons in the first 10-20 layers being more sensitive; and the transferability of safety neurons across multiple languages is investigated. Additionally, the authors propose the SN-Tune method, which exclusively fine-tunes safety neurons, achieving enhanced safety conditions by tuning only a small subset of parameters without catastrophic forgetting of general knowledge. The paper is easy to follow and well-written, with clear methodology explanations, comprehensive experimental results, and logical organization of findings.

**Strengths:**

1. Originality: The paper introduces an innovative approach to understanding LLM safety through the novel concept of "safety neurons." The clear definition and identification method for these neurons represents a significant departure from traditional safety alignment approaches, offering a fresh perspective on model safety mechanisms.

2. Experimental Rigor: The research demonstrates exceptional thoroughness in its experimental investigation, providing comprehensive analysis of safety neurons' characteristics, including their proportion (<1% of parameters), location sensitivity, robustness, and cross-lingual transferability. Each finding is well-supported by detailed empirical evidence.

3. Practical Impact: The proposed SN-Tune method offers a highly effective solution for enhancing model safety while maintaining general capabilities. Its ability to achieve significant safety improvements by tuning only a small subset of parameters makes it both efficient and practical for real-world applications, demonstrating immediate value for improving existing language models.

**Weaknesses:**

1. Theoretical Foundation: The paper lacks deeper theoretical analysis of observed phenomena. For example:
- The mechanism behind the "back-to-front deactivation" pattern remains unexplained, where safety breakdown only occurs after almost all safety neurons are deactivated
- This observation raises questions about whether the neurons in later layers are truly "safety neurons"
- The nature and characteristics of overlapping safety neurons across different languages deserve more theoretical investigation

2. Limited Analysis of Cross-lingual Safety Mechanisms:
- While the paper identifies low overlap (30%) between safety neurons across languages, it doesn't explore:
  * The characteristics of these overlapping neurons
  * Why certain neurons are shared across languages while others are language-specific
  * The potential universal principles of safety mechanisms across languages

**Questions:**

1. Position-Dependent Activation Patterns:
- How does the position of jailbreaking prompts affect safety neuron activation? Is there a correlation between prompt location and safety neuron response?
- Do safety neuron distributions shift in long-text scenarios？How does the context window length affect the stability of safety neurons?

2. Dynamic Nature of Safety Neurons during Fine-tuning:
- Can non-safety neurons transform into safety neurons during full-parameter fine-tuning? How does SN-Tune handle the potential emergence of new safety neurons?
- How can we track and verify the transformation of regular neurons into safety neurons?

---

> ### Author Response · Authors · 2024-11-20
>
> Dear Reviewer Uh2w,
>
> Thank you for your insightful reviews and comments. We appreciate the time and effort you have put into providing valuable feedback. We would like to address your concerns as follows:
>
> >  Suggestion #1 Theoretical foundation
>
> We appreciate your interest in the theoretical foundations and guarantees of our neuron detection method. Our work focuses on empirical analysis, defining, identifying, and verifying safety neurons, and exploring their features comprehensively. Building on our novel insights, we have developed two methods: SN-Tune for efficient safety alignment and RSN-Tune for robust safety alignment. While theoretical frameworks are valuable, they often involve numerous assumptions. We plan to explore these theoretical aspects in future work. Regarding the reviewer's concern about neurons in the final layers qualifying as "safety neurons," we note that these neurons are empirically identified as important in safety-related query handling. We find no compelling reason to exclude them from our definition of safety neurons merely because they can not independently manage safety mechanisms. We agree this is an interesting open question and have added it to our discussion of future work.
>
>
>
> > Suggestion #2 More Exploration
>
> We appreciate your thoughtful suggestions regarding additional aspects of safety neurons, including multilingual safety analysis, the impact of jailbreaking prompt positioning, and the effects of context window length. While these research directions would undoubtedly yield valuable insights, they extend beyond the current scope of our investigation. We plan to explore these promising directions in our future work. Furthermore, regarding safety neuron dynamics, SN-Tune exclusively modifies identified safety neurons while keeping other parameters fixed, which prevents non-safety neurons from transforming into safety neurons during fine-tuning.

---

### Official Review · Reviewer_9nQR · 2024-11-07

**Soundness:** 3
**Presentation:** 3
**Contribution:** 3
**Rating:** 6
**Confidence:** 2

**Summary:**

This paper presents an innovative method for effectively and efficiently detecting and fine-tuning "safety neurons," which comprise less than 1% of model parameters and are predominantly located in low-level self-attention layers. The authors conducted related experiments to verify that safety mechanism is resilient but breakable. Notably, the proposed tuning method $\texttt{SN-Tune}$ enhances model safety without sacrificing performance, significantly lowering harmful output scores in both instruction-tuned and base models. This approach also improves safety robustness during downstream fine-tuning tasks, such as GSM8K, by isolating safety neurons from foundational ones. Additionally, the authors explored the influence of the number of safety documents and performed grid searches over learning rates and training epochs in their ablation study.

**Strengths:**

* The paper is well-written and the idea is easy to be understood.
* Interesting verification of safety neurons. The authors used detailed experiments to verify that the identified safety neurons have a significant impact on the model's safety and found that these safety neurons are relatively few in number, primarily located in the self-attention layers of the first few blocks of the model.
* Extensive experiments. The authors validated the effectiveness of $\texttt{SN-Tune}$ on both instruction models and base models, as well as on downstream fune-tuning tasks.

**Weaknesses:**

* Although authors  empirically verify the identified safety neurons by their detection method have a significant impact on the model's safety, the proposed detection method lacks theoretical support for why it can identify neurons that have such a significant impact on safety.
* Why does the RHS of equations（4）（5） of the main paper  seem to be  independent of $l$ (the $l$-th neuron in the $i$-th layer)? They appear to be the same value for different $l$. Also in appendix, it is confusing why $h_{ffn}$ is a vector, could you help write the specific form? And how can you get (8) from (7) since x is a input for $W_{down}$  in (8) but not for $W_{down}$ in (7)?
* For downstream fine-tuning tasks, could you consider more datasets like Alpaca or Dolly to verify the effectiveness of $\texttt{RSN-Tune}$?

**Questions:**

* In table 4, why you used llama2-7B-chat (which is a safety-aligned model) and  Mistral-7B-Instruct-v0.2 (which is not a safety-aligned model) for downstream fine-tuning tasks ? How about use Mistral-7B models that have been safety-aligned by your proposed method $\texttt{SN-Tune}$?
* This paper [1] also proposes some detection methods to find safety neurons and perform operations on the identified neurons to reinforce the model's safety. Could the authors discuss more between this paper and  findings of your paper?
* There are several papers [1][2][3][4] also study on safety neuron/ safety mechanism/ backdoor, it will better if  the authors  could add some comments on them.

If the authors address most of my concerns, I would consider increasing the score.

[1] Wei, Boyi, et al. "Assessing the brittleness of safety alignment via pruning and low-rank modifications." arXiv preprint arXiv:2402.05162 (2024).

[2] Chen, Jianhui, et al. "Finding Safety Neurons in Large Language Models." arXiv preprint arXiv:2406.14144 (2024).

[3] Hsu, Chia-Yi, et al. "Safe LoRA: the Silver Lining of Reducing Safety Risks when Fine-tuning Large Language Models." arXiv preprint arXiv:2405.16833 (2024).

[4] Zhang, Zhengming, et al. "Neurotoxin: Durable backdoors in federated learning." International Conference on Machine Learning. PMLR, 2022.

---

> ### Author Response · Authors · 2024-11-20
>
> Dear Reviewer 9nQR,
>
> Thank you for your insightful reviews and comments. We appreciate the time and effort you have put into providing valuable feedback. We would like to address your concerns as follows:
>
> >  Concern #1 Theory guarantee
>
> We appreciate your interest in the theoretical foundations and guarantees of our neuron detection method. Our work focuses on empirical analysis, defining, identifying, and verifying safety neurons, and exploring their features comprehensively. Building on our novel insights, we have developed two methods: SN-Tune for efficient safety alignment and RSN-Tune for robust safety alignment. While theoretical frameworks are valuable, they often involve numerous assumptions. We plan to explore these theoretical aspects in future work.
>
>
>
> > Concern #2 Equations Clarification
>
> 1. Equations (4) and Equation (5) have the same value for different l in layer i.
>
> Yes, you are right. Equations (4) and (5) operate on all neurons l in layer i simultaneously, rather than processing each neuron individually. This parallel computation across neurons enables faster execution compared to sequential processing of each neuron.
>
> 2. $h_{ffn}$ and input of $W_{down}$ in Equation (7) and (8).
>
> We apologize for the typo. The dimension of $h_{ffn}$ should be $R^{l \times d_{inter}}$. I wrote $W_{down}(x)$ to indicate that the whole equation $|(h_{ffn}\cdot Mask[k]) W_{down}(x)|\_2$ involves $x$ as an input variable. This notation is not rigorous enough since $W_{down}$ does not directly take $x$ as input - the input information from $x$ is already contained in $h_{ffn}$. We will removed this notation in the updated version.
>
> > Concern #3 Implement RSN-Tune on Alpaca.
>
> Following your suggestion, we employ the setting of Table 3 in [1], where we train Llama2-7B-Chat and Mistral-7B-Instruct-v0.2 by Alpaca. Detailed results are as follows. We find that RSN-Tune can still enhance models’ safety robustness during Alpaca tuning.
>
> | Harmfulness              | Before | Original | SN-Tune | RSN-Tune |
> | :----------------------- | :----: | :------: | :-----: | :------: |
> | LLama2-7B-Chat           |  0.0   |   18.0   |  16.0   |   9.0    |
> | Mistral-7B-Instruct-v0.2 |  36.0  |   64.0   |  65.0   |   38.0   |
>
> >  Concern #4 Implement RSN-Tune on SN-Tune Mistral.
>
> Thanks for your suggestions. We respectfully disagree with the reviewer's assessment that Mistral-7B-Instruct-v0.2 is not safety-aligned. While a 36.0% harmful response rate indicates room for improvement, Mistral-7B-Instruct-v0.2 successfully refuses nearly two-thirds of harmful requests, demonstrating meaningful partial safety alignment. Furthermore, we conduct experiments on both well-aligned and partially aligned LLMs to verify the generalizability of RSN-Tune.
>
> Following your suggestion, we applied RSN-Tune to the SN-Tune-aligned Mistral-7B-v0.1 model. The results demonstrate that this combination enhances the model's safety mechanisms and makes them more resilient during downstream tuning. While the RSN-Tuned Mistral-7B-Instruct-v0.2 still shows a harmful behavior score of 41.0 after tuning on downstream task, our combined approach performs better due to SN-Tune's ability to partially separate safety neurons from general neurons—a capability not achievable through traditional safety alignment methods.
>
> |         | Before | Original | RSN-Tune |
> | ------- | ------ | -------- | -------- |
> | GSM8K   | 52.3   | 51.7     | 52.6     |
> | Harmful | 6.0    | 37.0     | 14.0     |

---

> ### Author Response · Authors · 2024-11-20
>
> > Concern #5 Comment on related work
>
> [2] employs various neuron detection methods to identify safety neurons and analyzes their brittleness through pruning. Their approach differs from ours in two key aspects. First, regarding neuron detection, [2] uses the SNIP score [3], which requires correct labels for gradient calculation, and implements [4], which enforces sparsity constraints like fixed sparse numbers in each column/row. In contrast, our method operates without requiring correct labels or uniform sparse parameters across columns/rows, enabling precise identification of 0.1% of parameters as safety neurons compared to their 3%, resulting in higher accuracy. Second, beyond analyzing safety neurons, we propose SN-Tune and RSN-Tune to enhance LLMs' safety alignment, an aspect not addressed in [2].
>
> Related works [5], [6], and [7] explore model safety mechanisms through different approaches. [5] limits its focus to the feed-forward layer, leading to imprecise detection and identifying 5% of parameters as safety neurons. [6] introduces additional structural elements, while [7] improves attacks by targeting parameters with minimal training changes—an approach that proves impractical due to the black-box nature of LLM training.
>
>
>
> [1] Qi, Xiangyu, et al. "Fine-tuning Aligned Language Models Compromises Safety, Even When Users Do Not Intend To!." International Conference on Learning Representations. 2024.
>
> [2] Wei, Boyi, et al. "Assessing the brittleness of safety alignment via pruning and low-rank modifications." arXiv preprint arXiv:2402.05162 (2024).
>
> [3] Lee, Namhoon, Thalaiyasingam Ajanthan, and Philip HS Torr. "Snip: Single-shot network pruning based on connection sensitivity." International Conference on Learning Representations (2022).
>
> [4] Sun, Mingjie, et al. "A Simple and Effective Pruning Approach for Large Language Models." The Twelfth International Conference on Learning Representations. (2024)
>
> [5] Chen, Jianhui, et al. "Finding Safety Neurons in Large Language Models." arXiv preprint arXiv:2406.14144 (2024).
>
> [6] Hsu, Chia-Yi, et al. "Safe LoRA: the Silver Lining of Reducing Safety Risks when Fine-tuning Large Language Models." arXiv preprint arXiv:2405.16833 (2024).
>
> [7] Zhang, Zhengming, et al. "Neurotoxin: Durable backdoors in federated learning." International Conference on Machine Learning. PMLR, 2022.

---

> ### Author Response · Authors · 2024-11-23
>
> Dear reviewer  9nQR,
>
> Thank you for your thoughtful comments and valuable feedback. We deeply appreciate the time and effort you have invested in reviewing our work.
>
> We have made our utmost effort to address your comments in the rebuttal. We clarified several details to address your converns and suggestions. If you feel that the clarifications align with your feedback, could you please consider raising your score in light of our efforts? Many thanks for your time; we are extremely grateful!
>
> Best regards, Authors

---

> > ### Comment · Reviewer_9nQR · 2024-11-24
> > **Thank you for your reply**
> >
> > Dear authors,
> >
> > Thanks for your detailed reply and extensive additional  experiments.
> >
> > >>Reply for $h_{ffn}$ and input of $W_{down}$ in Equation (7) and (8).
> >
> > Thanks for clarifying them, I think most of my concerns are addressed. I would also recommend the authors discussing more about the differences between your work and [1] in the final version,  it is interesting [1] also has these typos.
> >
> >
> > >> Implement RSN-Tune on Alpaca and SN-Tune Mistral.
> >
> > Thanks for your extensive experiments. I think my concerns are addressed.
> >
> > >> Comment on related work.
> >
> > Thanks for your additional discussion. It would be better if the authors could add some sentences to discuss them in the final version.
> >
> > I would correspondingly adjust my score. Thanks again for the detailed feedback!
> >
> > [1] Zhao, Yiran, et al. "How do Large Language Models Handle Multilingualism?." arXiv preprint arXiv:2402.18815 (2024).

---

> > > ### Author Response · Authors · 2024-11-24
> > >
> > > Dear Reviewer 9nQR,
> > >
> > > Thank you for reviewing our paper and considering our rebuttal. We greatly appreciate your recognition of our contribution and your valuable feedback. We will diligently incorporate your suggestions into the final version of our paper.
> > >
> > > Thank you once again for your time and expertise.
> > >
> > > Sincerely,
> > >
> > > The Authors

---

### Meta-Review · Area_Chair_CYnU · 2024-12-13

**Metareview:**

The recommendation is based on the reviewers' comments, the area chair's evaluation, and the author-reviewer discussion.

This paper studies safety alignment for LLMs by identifying and fine-tuning safety neurons. All reviewers find the studied setting meaningful and the results provide new insights. The authors’ rebuttal has successfully addressed the major concerns of reviewers.

In the post-rebuttal phase, the discussion was centered on the novelty and overlaps with existing works, especially the follow two paper:
-  "How do Large Language Models Handle Multilingualism?." NeurIPS (2024)."

The authors' method of identifying safety neurons borrows from the Parallel Language-specific Neuron Detection in this NeurIPS paper. Additionally, the idea of MWork also shared some similarities. Note that this NeurIPS paper did not study safety, so the use case considered in this paper is new.

- "Assessing the brittleness of safety alignment via pruning and low-rank modifications." ICML (2024).

The idea of locating and fine-tuning safety-critical neurons is studied in this ICML paper.

Overall, this is a borderline paper. The novelty and originality are not outstanding because of the overlap and similarity to existing works. On the other hand, the experiments are thorough and the empricla results are convincing. The studied topic is also timely and important. I recommend acceptance of this submission. I also expect the authors to include the new results and suggested changes during the rebuttal phase to the final version.

**Additional Comments On Reviewer Discussion:**

In the post-rebuttal phase, the discussion was centered on the novelty and overlaps with existing works, especially the follow two paper:
-  "How do Large Language Models Handle Multilingualism?." NeurIPS (2024)."

The authors' method of identifying safety neurons borrows from the Parallel Language-specific Neuron Detection in this NeurIPS paper. Additionally, the idea of MWork also shared some similarities. Note that this NeurIPS paper did not study safety, so the use case considered in this paper is new.

- "Assessing the brittleness of safety alignment via pruning and low-rank modifications." ICML (2024).

The idea of locating and fine-tuning safety-critical neurons is studied in this ICML paper.

Overall, this is a borderline paper. The novelty and originality are not outstanding because of the overlap and similarity to existing works. On the other hand, the experiments are thorough and the empricla results are convincing.

---

### Decision · Program_Chairs · 2025-01-22

Accept (Poster)